# A Pilot Randomised Controlled Trial on the Effectiveness of an Anti-Stress Ball Technique for Pain Reduction during Vascular Access Cannulation in Haemodialysis Patients

**Magda Dinis [1] and Joana Pereira Sousa [2,*]**

[1]  School of Health Sciences, Polytechnic of Leiria, Diaverum, F3080-847 igueira da Foz, Portugal
[2]  School of Health Sciences, Polytechnic of Leiria, Center for Innovative Care and Health Technology, ciTechCare, 2411-901 Leiria, Portugal
*  Correspondence: joana.sousa@ipleiria.pt

**Abstract:** Background: Pain in patients under haemodialysis affects the quality of life of chronic renal patients. Distraction has been effective in controlling pain induced by the insertion of needles. Once applied adequately, distraction promotes endorphin release, with efficacy in acute pain management. This study evaluates pain perception while puncturing the haemodialysis device, using an anti-stress ball as a distraction strategy. Methods: This study is a retrospectively registered pilot randomised controlled trial of 47 chronic renal patients undergoing regular haemodialysis programs in a dialysis unit, in the central region of Portugal. Patients were randomly allocated into control and intervention groups. The intervention group received an anti-stress ball on the opposite limb of the vascular access, while the control group underwent pain evaluation without any intervention. Pain was monitored using a numeric pain scale. The study aimed to evaluate pain during vascular access cannulation and explore the potential benefits of using an anti-stress ball to reduce this pain. Results: Patients in the intervention group experienced significantly lower pain scores ($-1.23$; $p < 0.05$) during vascular access cannulation compared to the control group ($-0.51$). Conclusion: Distraction through an anti-stress ball reduces the perception of pain experienced by the chronic renal patients undergoing haemodialysis. However, the trial's retrospective registration may introduce a risk of selective-outcome reporting. Further research with prospectively registered trials is recommended to validate the findings.

**Keywords:** renal dialysis; arteriovenous fistula; renal insufficiency chronic; quality of life; distraction

## 1. Introduction

Kidneys maintain fluid, electrolyte, and metabolic-acid balance [1]. Kidney disease is referred to as a kidney abnormality. When the kidney reduces its normal function, less than 30% of normal activity is related to a long-term condition [2,3]. This affects millions of people worldwide. In severe states where the kidney deteriorates to a serious level, the kidney can no longer work properly, and therefore, people need to participate in a haemodialysis (HD) programme [2,3]. In haemodialysis, blood is filtered by an external machine, done about three times a week, and the access is gained through intermittent vascular cannulation [3]. The main cannulation techniques are rope-ladder, buttonhole, and puncture [4,5]. It is estimated that an average haemodialysis patient undergoes 312 cannulations per year [6]. This procedure harms the patient and provokes anxiety due to fear, pain, and distress [5]. HD nurses need good cannulation skills to minimise anxiety and pain. Pain in HD patients ranges between 12% and 80%, and is a daily challenge for nursing care [6].

Raghibi et al. (2018) showed that distraction effectively controls pain resulting from needle insertion. Distraction aims to divert concentration away from the painful stimulus and can reduce anxiety, fear, and pain intensity itself. When properly performed, distraction promotes endorphin release, which is effective in acute pain management [7].

Three strategies have been identified in reducing pain during cannulation of the arteriovenous fistula for haemodialysis: (1) puncture technique, indicating the buttonhole technique as the one that translates a lower level of pain during cannulation; (2) use of non-pharmacological therapies, such as transcutaneous electrical stimulation, placing the users' feet in hot water ten minutes before puncture, cryotherapy at the puncture site, acupuncture, audio-visual distraction techniques, and aromatherapy with lavender essence, or thermotherapy of the vascular access (of these, cryotherapy was identified as the technique that produced the best results in reducing the pain at the puncture of the vascular access); and (3) application of local anaesthetic before puncturing the venous access, identifying creams based on lidocaine and prilocaine (EMLA®) as ones that had the most effective results in reducing pain [8–10].

Pharmacological and non-pharmacological methods are considered to relieve pain in cannulating vascular access for haemodialysis. In contrast, non-pharmacological means are cheaper, more accessible, and easily applicable. Pain during cannulation of vascular access is a problem that lacks adequate intervention programs based on non-pharmacological methods for pain relief [11].

The primary objective of this pilot study is to assess the feasibility of implementing an anti-stress ball technique in a haemodialysis setting, evaluating recruitment rates, retention rates, and participant compliance with the intervention. The secondary objective is to provide preliminary evidence on the effectiveness of the anti-stress ball technique in reducing pain perception during vascular access cannulation for haemodialysis patients.

## 2. Materials and Methods

The pilot study used a randomised controlled trial with a parallel design, where participants were randomly assigned to either a control group or an intervention group for 1:1 randomisation, and the participants were blinded.

### 2.1. Participants

The study used a convenience sample composed of patients undergoing regular haemodialysis programmes in a dialysis unit, in the central region of Portugal. Patients were included if they were adults over 18 years old, able to read and write, and had vascular access by a fistula or arteriovenous prosthesis in use for more than one month, cannulated with a 15 G needle. Patients under 18 years, who could not read and write, used topical anaesthetics before treatment, had vascular access for less than a month, or were cannulated with needles smaller or larger than 15 G, were excluded from the study.

### 2.2. Intervention

The intervention group received an anti-stress ball on the opposite limb of the vascular access. They were instructed to squeeze and repeatedly release while looking away from it, until the cannulation process was completed. The control group only underwent pain evaluation during vascular access cannulation.

### 2.3. Outcomes

The primary outcome was the level of pain experienced by patients during vascular access cannulation. Pain was measured using a numeric pain scale at the end of the procedure. To establish the patients' pain profiles before the intervention, the pain was evaluated in the first two weeks of the study start, which corresponded to six moments (three times a week), using the numeric pain scale for all patients. This initial pain assessment was conducted without any intervention, as pain was not previously evaluated.

### 2.4. Sample Size

The required sample size was calculated based on a 95% confidence level, a margin of error of 5%, a 1.2% proportion of the condition, and a population size of 10,000,000 people.

The calculated sample size was 35 participants. However, the study used a convenience sample, which included all eligible participants ($n = 47$).

### 2.5. Randomisation

Randomisation was achieved using a quasi-randomisation method based on patient presence in the dialysis unit. Patients who were present on day one were allocated to the intervention group, and those present on day two were allocated to the control group. This method resulted in a 1:1 allocation ratio between the two groups. It is important to note that this randomisation method may introduce potential biases and confounding factors, as it does not fully ensure the equal distribution of known and unknown prognostic factors across the groups.

To establish the patients' pain profiles before the intervention, the pain was evaluated in the first two weeks of the study start, which corresponded to six moments (three times a week), using the numeric pain scale for all patients. This initial pain assessment was conducted without any intervention. Overall, the study aimed to evaluate pain during vascular access cannulation and explore the potential benefits of using an anti-stress ball to reduce this pain.

When starting the intervention, the control group continued pain assessment without any intervention, while the intervention group received the distraction technique in addition to pain assessment by scale. The study was divided into two parts, with four weeks designated for its application (Figure 1). Enrolment began in July 2022, and the study concluded in September 2022.

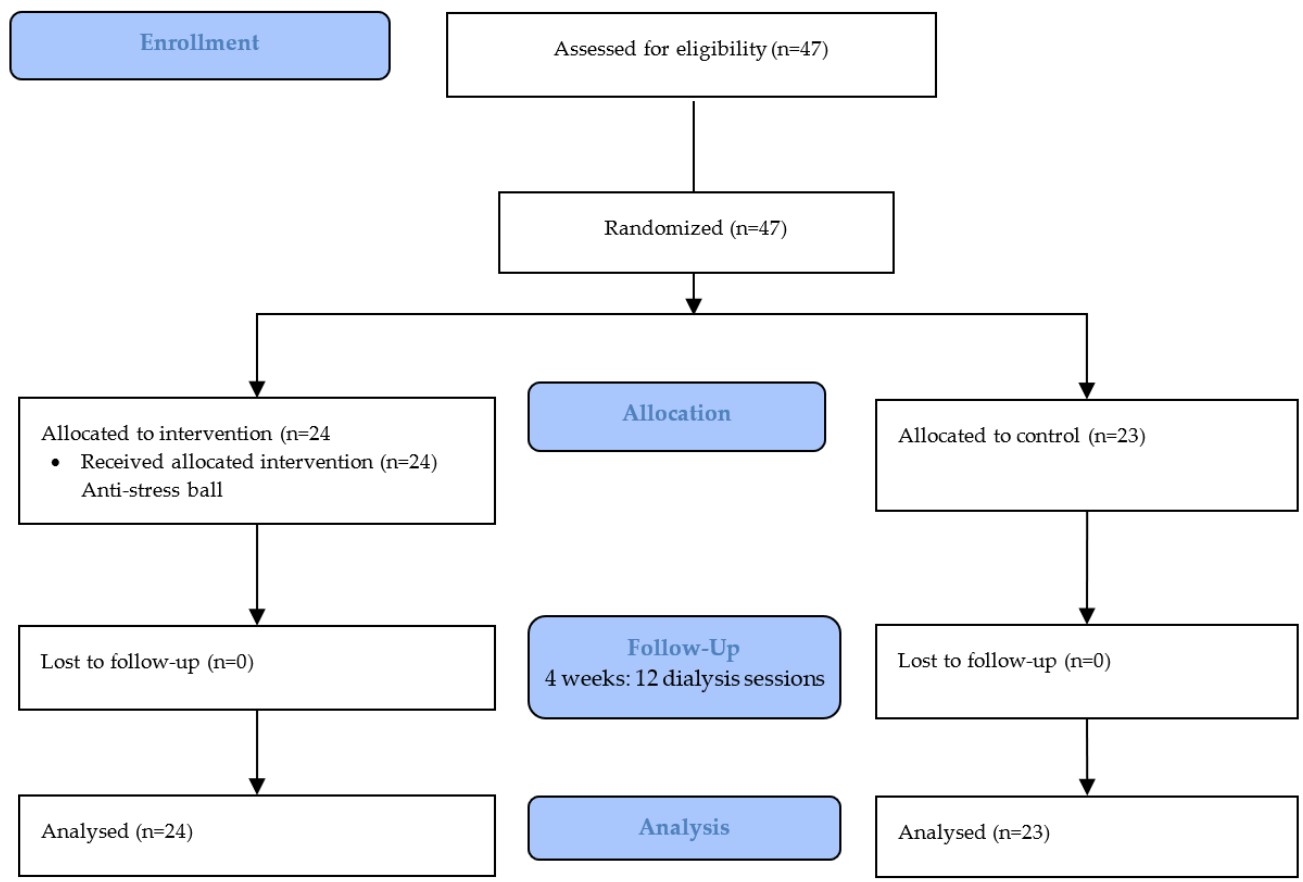

**Figure 1.** RCT CONSORT flowchart.

### 2.6. Harms

No adverse events or harms were reported by the participants during the study. The use of an anti-stress ball was a non-invasive intervention, and no negative effects were

observed. The potential for harm in the future full-scale trial will be carefully monitored and reported.

*2.7. Statistical Methods*

Data were collected using a form divided into three parts: sociodemographic characterisation, clinical characterisation, and assessment of pain perception using a numerical 0–10 scale.

Data were analysed using intention-to-treat principles. Descriptive statistics were used to summarize demographic and clinical characteristics. The primary outcome of the pain level was analysed using an independent samples *t*-test to compare differences in pain scores between the intervention and control groups. Paired *t*-tests were employed to compare pain scores within each group before and after the intervention. The level of significance was set at $p < 0.05$. The IBM Statistical Package for the Social Science (SPSS®) v.27 software was used for data analysis.

*2.8. Ethics*

Ethical considerations were carefully addressed throughout the pilot study (DIAV-105 on 6 June 2022). All participants provided written informed consent before participating in the study, and their confidentiality was maintained by anonymizing data and ensuring that no personally identifiable information was disclosed. The study protocol was approved by the relevant institutional review board, and all research activities were conducted in accordance with the principles of the Declaration of Helsinki and local ethical guidelines.

The pilot trial was registered with the RCT—NCT05729113, and the study protocol is available upon request. No significant changes were made to the protocol during the study. Any minor deviations from the protocol were documented and will be considered when designing the future full-scale trial.

**3. Results**

According to the results obtained in Table 1, the study sample comprised 47 participants, of which 23 belonged to the control group (48.94%) and 24 to the intervention group (51.06%). The participants were mostly male, with 56.5% in the control group and 58.3% in the intervention group. The mean age of the sample in the control group was 65.6 (SD = 13.8) years, and the mean age of the intervention group was 69.4 (SD = 11.9) years.

In the control group, 52.5% lived in a village, 26.1% lived in a town, and 21.7% lived in a city. For the intervention group, 37.5% lived in a village, 29.2% lived in a town, and 33.3% lived in a city. Most of the sample had completed primary education (56.5% in the control group and 50.0% in the intervention group), and the meantime on dialysis was 65.96 (SD = 73.12) months in the control group and 79.21 (SD = 85.84) months in the intervention group.

Regarding the type of vascular access, 95.7% of the control group used arteriovenous fistula (AVF) and 4.3% of prosthetic haemodialysis access (PHE). The time of use of vascular access, as shown in Table 1, was 40.09 (SD = 36.13) months in the control group and 54.83 (50.26) months in the intervention group.

The vascular accesses were located in the wrist (13%), forearm (13%), elbow (39.1%), and arm (8%) in the control group. In the intervention group, 20.8% of the access was in the wrist, 20.8% in the forearm, 25% in the elbow, and 8% in the arm.

Table 1 shows that, in the control group, 43.5% of the accesses had been subject to intervention and 56.5% had not, with a mean time after the intervention of 4.40 (SD = 4.30) months. In the intervention group, 25% of the accesses had been subject to intervention, and 75% had not, with a mean time after the intervention of 17.83 (SD = 15.04) months.

**Table 1.** Participant characteristics.

| Variable | | Control | | Intervention | | Sig. |
|---|---|---|---|---|---|---|
| | | **N** | **%** | **N** | **%** | |
| Sex | Male | 13 | 56.5 | 14 | 58.3 | 0.900 [a] |
| | Female | 10 | 43.5 | 10 | 41.7 | |
| Residency | Village | 12 | 52.2 | 9 | 37.5 | |
| | Town | 6 | 26.1 | 7 | 29.2 | |
| | City | 5 | 21.7 | 8 | 33.3 | |
| Education | Primary School | 13 | 56.5 | 12 | 50.0 | |
| | Elementary School | 5 | 21.7 | 4 | 16.7 | |
| | Secondary Education | 1 | 4.3 | 5 | 20.8 | |
| | Higher Education | 4 | 17.4 | 3 | 12.5 | |
| Type of vascular access | AVF | 22 | 95.7 | 22 | 91.7 | 0.516 [b] |
| | PHE | 1 | 4.3 | 2 | 8.3 | |
| Vascular access location | Wrist | 3 | 13.0 | 5 | 20.8 | 0.664 [a] |
| | Forearm | 3 | 13.0 | 5 | 20.8 | |
| | Elbow | 9 | 39.1 | 6 | 25.0 | |
| | Arm | 8 | 34.8 | 8 | 33.3 | |
| Vascular access intervention | Yes | 10 | 43.5 | 6 | 25.0 | |
| | No | 13 | 56.5 | 18 | 75.0 | |
| | | Mean | SD | Mean | SD | Sig |
| Age | | 65.57 | 13.77 | 69.42 | 11.87 | 0.309 [c] |
| Time of dialysis | | 65.96 | 73.12 | 79.21 | 85.84 | |
| Time of vascular access | | 40.09 | 36.13 | 54.83 | 50.26 | |
| Time after vascular access intervention | | 4.40 | 4.30 | 17.83 | 15.04 | |

[a] Chi-square; [b] Fisher's exact test; [c] Student's *t* test.

As for the mean pain level (Table 2), we found that, in the control group, participants verbalised a mean pain level of 2.08 (SD = 1.47) in the first six assessments (Before) and that in the following six assessments (After), participants verbalised a mean pain level of 1.56 (SD = 1.17), which translates into a decrease in the mean pain level of 0.51 (1). In turn, the intervention group had a mean pain level of 3.01 (SD = 1.12) in the first six moments of assessment (Before) and had a mean pain level of 1.78 (SD = 1.15) in the following six moments, which corresponds to the moment of intervention in this group (After intervention), which translates a decrease in the mean pain level of 1.23 (SD = 1.3).

**Table 2.** Pain evaluation.

| | Control M (SD) | Intervention M (SD) | *p* Levene | *t* | *p* | D |
|---|---|---|---|---|---|---|
| Mean Before | 2.08 (1.47) | 3.01 (1.12) | 0.309 | −2.43 | 0.019 | −0.709 |
| Mean After | 1.56 (1.17) | 1.78 (1.15) | 0.930 | −0.63 | 0.532 | −0.184 |
| Mean Difference | −0.51 (1) | −1.23 (1.3) | 0.180 | 2.11 | 0.020 | 0.615 |

A Student's *t*-test showed statistically significant differences between the two groups concerning the reduction in mean pain levels before and after the intervention ($p < 0.05$). Statistically significant differences were observed between both groups concerning the mean pain before the intervention ($p < 0.05$).

Table 3 shows that in the control group, 65.2% of participants had their pain level improved over the observation period, 13.0% had their pain level maintained, and 21.7% had their pain level worsened. In the intervention group, 75% of the participants had

their pain level improved, 20.8% reported worsening, and only 4.2% reported maintaining pain level.

**Table 3.** Cannulation of vascular access pain perception.

| Pain Perception | Control *n* (%) | Intervention *n* (%) |
| --- | --- | --- |
| Better | 15 (65.2) | 18 (75.0) |
| The same | 3 (13.0) | 1 (4.2) |
| Worst | 5 (21.7) | 5 (20.8) |

## 4. Discussion

Pain at the time of cannulation for vascular access during haemodialysis is a common concern for chronic renal patients who undergo this procedure three times a week. This procedure involves the insertion of needles into the patient's arteriovenous fistula or graft, which can cause discomfort and pain. Patients may also experience anxiety and fear of the procedure, which can exacerbate their pain perception. Therefore, minimizing pain during this procedure is crucial to improving patients' quality of life and promoting compliance with the treatment regimen [12,13].

The study's results showed that the mean age of patients in both the control and intervention groups was consistent with the records of the Portuguese Nephrology Society, which indicated that the majority of patients undergoing renal function replacement treatment in Portugal are elderly, with a mean age of 67.5 years [14]. In the study, the control group had a mean age of 65.6 years, and the intervention group had a mean age of 69.4 years, indicating that the patients in the study were representative of the broader population of haemodialysis patients in Portugal [12].

Furthermore, the study found that the gender distribution of the patients was consistent with the SPN data, with males comprising the majority of patients in both the control and intervention groups. Specifically, the control group had 57% males and 44% females, while the intervention group had 58% males and 42% females. The study did not reveal any statistically significant differences related to gender or time under haemodialysis treatment, which is consistent with the findings of previous studies [7,15].

Pain associated with cannulation is a common and significant problem in haemodialysis patients. Cannulation is the process of accessing the patient's bloodstream through a vascular access point to remove blood and filter it through the dialysis machine. The most common types of vascular access points are arteriovenous fistula (AVF), arteriovenous graft (AVG), and central venous catheter (CVC). However, the cannulation process can be painful and uncomfortable, and patients often report anxiety and fear of pain associated with the procedure. Therefore, it is essential to manage pain during the cannulation process to improve the patient's experience and adherence to the treatment. Different pain management strategies can be used, including topical anaesthetics, distraction techniques, and psychological support. Furthermore, healthcare providers should regularly assess and monitor pain levels in haemodialysis patients to ensure optimal pain management.

These findings suggest that pain associated with vascular access cannulation is a common concern for patients undergoing haemodialysis. However, the severity of pain reported can vary across studies [6,11,13–15]. In the present study, the mean pain level reported by the participants was consistent with the mild-to-moderate pain reported in previous research [13,14]. It is worth noting that the intervention group in the current study reported a slightly higher mean pain level before the intervention than the control group, which could indicate a greater need for pain management interventions in this group. Additionally, the study referenced in the last sentence found that a significant percentage of patients experienced severe pain during cannulation, highlighting the importance of addressing this issue in clinical practice [15].

Kaza et al. (2014) obtained statistically insignificant results for time on dialysis ($p = 0.690$), type of vascular access ($p = 0.563$), and duration of vascular access ($p = 0.806$) in

relation to pain perception during vascular access cannulation, which is consistent with our findings [16]. However, they did find a significant reduction in pain associated with the location of the vascular access ($p < 0.05$). In our study, we observed a reduction in the mean difference in pain before and after the intervention in both groups ($p < 0.05$). However, we did not observe any significant differences in pain perception between the different puncture sites ($p > 0.05$). Variables such as time on dialysis, type of vascular access, and duration of vascular access, also showed no statistical significance in reducing pain during vascular access cannulation for haemodialysis in either group ($p > 0.01$).

Regarding the distraction effect on decreasing pain intensity at the time of needle insertion in the arteriovenous fistula for haemodialysis, Raghibi et al. (2018) concluded that distraction reduced the intensity of pain during needle insertion for haemodialysis ($p < 0.002$) [7], which is in line with the results of our study, where there was a higher reduction in the intervention group in the difference of mean pain scores before and after intervention ($p < 0.05$). A study by Nasirzadeh and collaborators, in 2019, compared two distraction techniques (guided visualisation and virtual reality) as non-pharmacological strategies. Both techniques positively affected decreasing the mean pain level during arteriovenous fistula cannulation. The group that tested the effect of virtual reality had a significantly lower mean pain level compared to the guided visualisation group ($p < 0.001$) [17]. These results suggest a positive effect of distraction in reducing pain during vascular access cannulation for haemodialysis patients. When the effect of the association of two non-pharmacological techniques (distraction and cryotherapy) was analysed regarding the effect of an isolated technique (cryotherapy) on the reduction of the mean level of pain at the time of cannulation of the arteriovenous fistula for haemodialysis, it was seen that the association between distraction and cryotherapy was more effective in reducing the mean level of pain than cryotherapy per se ($p < 0.05$) [18].

This study assessed the effectiveness of using an anti-stress ball as a distraction technique to reduce pain perception during vascular access cannulation for haemodialysis. Both control and intervention groups showed decreased pain perception between the first and second part of the study. However, the intervention group showed a greater difference in pain perception than the control group, with statistically significant differences. Participants in the intervention group reported the effectiveness of the distraction technique.

Although the study findings provide valuable insights into the effectiveness of using an anti-stress ball as a distraction technique in reducing pain perception during vascular access cannulation for haemodialysis patients, there are several limitations to consider. First, the study had a small sample size, which limits the generalizability of the results. Second, the study was conducted in a single centre, which may not represent the broader population of haemodialysis patients in Portugal. Third, the study did not account for other factors that may contribute to pain perception, such as anxiety levels, individual pain tolerance, and the presence of comorbidities. Fourth, the study did not assess the long-term effects of the intervention, and it is unclear whether the intervention can sustainably reduce pain perception during vascular access cannulation over time. Lastly, the retrospective registration of the trial may introduce a risk of selective-outcome reporting.

Further research is needed to explore the effectiveness of other non-pharmacological strategies in reducing pain perception during vascular access cannulation for haemodialysis. The lack of studies reporting on this issue highlights the need for more research in this field, including prospectively registered trials with rigorous randomisation methods.

## 5. Conclusions

The study suggests that the anti-stress ball technique can be a useful intervention for pain reduction during vascular access cannulation for haemodialysis. The low cost of the anti-stress ball technique makes it a recommended intervention for healthcare professionals to consider when caring for haemodialysis patients. However, the trial's retrospective registration may introduce a risk of selective-outcome reporting.

Further research is needed to explore the effectiveness of other non-pharmacological strategies in reducing pain perception during vascular access cannulation for haemodialysis, with larger sample sizes and longer data collection periods, and multiple centres. Additionally, exploring other non-pharmacological strategies for pain reduction in haemodialysis patients could provide valuable insights and potentially improve patient outcomes. Prospective trials with rigorous randomisation methods are recommended to validate the findings of this study and advance the understanding of pain management in this patient population.

**Author Contributions:** Conceptualization, M.D. and J.P.S.; methodology, M.D. and J.P.S.; software, M.D. and J.P.S.; validation, M.D. and J.P.S.; formal analysis, J.P.S.; investigation, M.D.; resources, M.D.; data curation, M.D. and J.P.S.; writing—original draft preparation, M.D.; writing—review and editing, J.P.S.; visualization, J.P.S.; supervision, J.P.S.; project administration, J.P.S. All authors have read and agreed to the published version of the manuscript.

**Funding:** This research received no external funding.

**Institutional Review Board Statement:** The study was conducted according to the Declaration of Helsinki and approved by the Ethics Committee of Equipa de Investigação e Desenvolvimento Empresarial da Diaverum (DIAV-105 on 6 June 2022). This trial was retrospectively registered on the 6 February 2023, registration number: NCT05729113.

**Informed Consent Statement:** Informed consent was obtained from all subjects involved in the study.

**Data Availability Statement:** The datasets generated and analysed during the current study are not publicly available due to the need to protect patient privacy and comply with ethical guidelines. However, summary statistics and key findings are presented in the manuscript. Requests for access to the de-identified data for research purposes may be considered upon reasonable request.

**Conflicts of Interest:** The authors declare no conflict of interest.

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
