# Peer review of "A Pilot Randomised Controlled Trial on the Effectiveness of an Anti-Stress Ball Technique for Pain Reduction during Vascular Access Cannulation in Haemodialysis Patients"

_nursrep, doi:10.3390/nursrep13020064_

Round 1

Reviewer 1 Report

Dear authors,

As a pediatric nephrologist with 24 years of experience in pediatric dialysis, I was pleased to be contacted to review your article. Pain perception by the chronic kidney patient at the time of vascular access cannulation is one of the most important issues in chronic dialysis. I want to congratulate you on your article, which, even if it reflects the situation in a small dialysis center, can represent a starting point for a multicenter study. Considering that chronic end-stage kidney disease cannot be cured (except by transplantation), improving the quality of life becomes the essential objective of health care.

As for the article, it met the requirements of a randomized controlled trial. I would recommend the following:

- improving the introduction by describing the problem (CKD, dialysis, intermittent vascular catheterization)

- the paragraph between lines 34-39 could be moved to discussions, there is much more to be said in that section

- paragraph 63 - 66 could be moved to the Materials and method section

The results section is well organized, clearly laid out

The Discussion section is insufficient in my opinion. Much more could be discussed on account of the results obtained and which are quite systematized.

Also, the conclusions are more discussions. I really suggest moving the data to the discussion section and drawing up specific conclusions.

More references could improve the quality of the text

Good luck in your work!

With coonsideration

Author Response

Reviewer 1

Thank you for your comments.

We have considered your suggestions and made changes, tracked in the article in blue colour:

- improving the introduction by describing the problem (CKD, dialysis, intermittent vascular catheterization) – added information in the Introduction

- the paragraph between lines 34-39 could be moved to discussions, there is much more to be said in that section – it has been moved to discussion

- paragraph 63 - 66 could be moved to the Materials and method section – it has been moved to Materials and Methods

The results section is well organized, clearly laid out – thank you

The Discussion section is insufficient in my opinion. Much more could be discussed on account of the results obtained and which are quite systematized – The Discussion section has been improved. However, because of lack of studies on this area (pain in the vascular access cannulation)

Also, the conclusions are more discussions. I really suggest moving the data to the discussion section and drawing up specific conclusions. – The conclusion has been improved.

More references could improve the quality of the text – references were added.

Kind regards,

The authors

Reviewer 2 Report

This nicely written paper analyzes the application of an anti-stress ball on pain perception during vascualr access cannulation.

In the introduction section I would add some more literature references especially regarding the linesa 45 to 55.

In the materials and methods section the type of numerical scale, i suppose 0-10 was used, should be stated.

In the result section the values in the brackets shloud be explained, for example line 138: "in the control group was 65.6 (13.8).." I supposte the value 13.8 corresponding to the standard deviation. In Table 2: "M (DP)" should be explained.

The chapter regarding the "theoretical model" should be cut away (lines 202 to 229). The "theoretical model" does not support the previously presesented results and is generating only "confusion".

In the discussion section the second part of the last sentence "...although, in our case, the was no association of two non-pharmacological strategies." (lines 283-286) can be eliminated.

Author Response

Reviewer 2

Thank you for your comments.

We have considered your suggestions and made changes, tracked in the article in blue colour:

In the introduction section I would add some more literature references especially regarding the linesa 45 to 55. – references were added in this section.

In the materials and methods section the type of numerical scale, i suppose 0-10 was used, should be stated. - In the materials and methods section, the pain scale used was the analogic pain scale.

In the result section the values in the brackets shloud be explained, for example line 138: "in the control group was 65.6 (13.8).." I supposte the value 13.8 corresponding to the standard deviation. In Table 2: "M (DP)" should be explained. - DP was added and corrected. Thank you for noticing it.

The chapter regarding the "theoretical model" should be cut away (lines 202 to 229). The "theoretical model" does not support the previously presented results and is generating only "confusion". - The theoretical model was removed from the manuscript.

In the discussion section the second part of the last sentence "...although, in our case, the was no association of two non-pharmacological strategies." (lines 283-286) can be eliminated. – these lines were removed

Kind regards,

The authors

Reviewer 3 Report

This study investigated whether an anti-stress ball usage reduces pain experienced by chronic renal failure patients undergoing hemodialysis during vascular access cannulation. I have several comments for the authors.

1. It seems that the patients in the control group didn’t receive any intervention to reduce the pain or discomfort during the procedure. If so, the study design seems inappropriate. Please clarify what intervention the control group received and if there was any plan for the patients in the control group if they expressed any pain during the study.

2. How was the sample size determined in this study?

3. Patients with chronic renal failure usually have other co-existing diseases, such as cardiovascular problems. Were the hemodynamics monitored when the vascular access cannulation was performed? If so, please add the relevant data in the manuscript.

Author Response

Reviewer 3

Thank you for your comments.

We have considered your suggestions and made changes, tracked in the article in blue colour:

  1. It seems that the patients in the control group didn’t receive any intervention to reduce the pain or discomfort during the procedure. If so, the study design seems inappropriate. Please clarify what intervention the control group received and if there was any plan for the patients in the control group if they expressed any pain during the study. - Please see the material and method section where information was organised about the design of intervention and control groups.

  1. How was the sample size determined in this study? - Sample size was not determined. A convenience sample was used.

  1. Patients with chronic renal failure usually have other co-existing diseases, such as cardiovascular problems. Were the hemodynamics monitored when the vascular access cannulation was performed? If so, please add the relevant data in the manuscript. - Our study aimed to assess pain at the time of vascular cannulation. We didn't evaluate co-existing diseases, as they were irrelevant to our study.

Kind regards,

The authors

Round 2

Reviewer 1 Report

Dear authors,

thank you for your reply!

 The work is significantly improved.

I would recommend the following as minor changes, before accepting the work:

From next two sentences in red, I would summarize a single conclusion regarding the advantage of using the stress ball during cannulation for hemodialysis, due to the accessibility and low cost of this method.

The low cost of the anti-stress ball technique makes it a recommended intervention for healthcare professionals to consider when caring for hemodialysis patients.

The low cost of the anti-stress ball technique makes it a recommended intervention for pain reduction during vascular access cannulation for hemodialysis.

The conclusions could be reduced to the following phrases:

The study suggests that the anti-stress ball technique can be a useful intervention for pain reduction during vascular access cannulation for hemodialysis.

The low cost of antistres ball….

Further research is needed to explore the effectiveness of other non-pharmacological strategies in reducing pain perception during vascular access cannulation for hemodialysis, with more participants and longer data collection periods.

The paragraphs:

-          274-283 - In addition to the effectiveness……..more research in this field.

-          289-295 - This study assessed the effectiveness …….. of the distraction technique.

-          296-297- Despite the positive findings,…….. the small sample size

-          300-301 – The lack of studies reporting this issue….. studies in this field

could be moved to Discussions, because they talk about the limitations of the study.

With consideration

Author Response

Dear reviewer.

Thank you for your you comments.

We have included limitations in the discussion section and reduced the Conclusion section. Changes are in orange color.

Kind regards,

The authors

Reviewer 3 Report

I appreciate the authors’ response. No further comments.

Author Response

Dear reviewer.

We thank you for your support is the revision of this manuscript.

Kind regards,

The authors,